# High association of COVID-19 severity with poor gut health score in Lebanese patients

Imad Al Kassaa[1,2]*, Sarah El Omari[3], Nada Abbas[4], Nicolas Papon[5], Djamel Drider[6], Issmat I. Kassem[7], Marwan Osman[8]*

1 Faculty of Public Health, Lebanese University, Beirut, Lebanon, 2 Doctoral School of Science and Technology, Lebanese University, Beirut, Lebanon, 3 Department of Epidemiology and Population Health, Faculty of Health Sciences, American University of Beirut, Beirut, Lebanon, 4 Department of Health Management and Policy, Faculty of Health Sciences, American University of Beirut, Beirut, Lebanon, 5 Univ Angers, Univ Brest, GEIHP, SFR ICAT, Angers, France, 6 UMR Transfrontalière BioEcoAgro1158, Univ. Lille, INRAE, Univ. Liège, UPJV, YNCREA, Univ. Artois, Univ. Littoral Côte d'Opale, ICV—Institut Charles Viollette, Lille, France, 7 Center for Food Safety and Department of Food Science and Technology, University of Georgia, Griffin, GA, United States of America, 8 Department of Population Medicine and Diagnostic Sciences, College of Veterinary Medicine, Cornell University, Ithaca, NY, United States of America

* imad.kassaa@ul.edu.lb (IAK); mo368@cornell.edu, marwan.osman@outlook.com (MO)

## Abstract

### Background

Coronavirus disease 2019 (COVID-19) has affected millions of lives globally. However, the disease has presented more extreme challenges for developing countries that are experiencing economic crises. Studies on COVID-19 symptoms and gut health are scarce and have not fully analyzed possible associations between gut health and disease patho-physiology. Therefore, this study aimed to demonstrate a potential association between gut health and COVID-19 severity in the Lebanese community, which has been experiencing a severe economic crisis.

### Methods

This cross-sectional study investigated SARS-CoV-2 PCR-positive Lebanese patients. Participants were interviewed and gut health, COVID-19 symptoms, and different metrics were analyzed using simple and multiple logistic regression models.

### Results

Analysis of the data showed that 25% of participants were asymptomatic, while an equal proportion experienced severe symptoms, including dyspnea (22.7%), oxygen need (7.5%), and hospitalization (3.1%). The mean age of the participants was 38.3 ±0.8 years, and the majority were males (63.9%), married (68.2%), and currently employed (66.7%). A negative correlation was found between gut health score and COVID-19 symptoms (Kendall's tau-b = -0.153, P = 0.004); indicating that low gut health was associated with more severe COVID-19 cases. Additionally, participants who reported unhealthy food intake were more likely to experience severe symptoms (Kendall's tau-b = 0.118, P = 0.049). When all items were taken into consideration, multiple ordinal logistic regression models showed a

**Data Availability Statement:** All relevant data are within the manuscript and its Supporting Information files.

**Funding:** The authors received no specific funding for this work

**Competing interests:** The authors have declared that no competing interests exist.

significant association between COVID-19 symptoms and each of the following variables: working status, flu-like illness episodes, and gut health score. COVID-19 severe symptoms were more common among patients having poor gut health scores (OR:1.31, 95%CI:1.07–1.61; P = 0.008), experiencing more than one episode of flu-like illness per year (OR:2.85, 95%CI:1.58–5.15; P = 0.001), and owning a job (OR:2.00, 95%CI:1.1–3.65; P = 0.023).

## Conclusions

To our knowledge, this is the first study that showed the impact of gut health and exposure to respiratory viruses on COVID-19 severity in Lebanon. These findings can facilitate combating the pandemic in Lebanon.

## Introduction

As of September 21[st], 2021, the coronavirus disease 2019 (COVID-19) has affected 230 million individuals around the world and claimed the lives of 4.7 million people [1]. COVID-19 can cause symptoms that range from mild to life-threatening. The etiologic agent of COVID-19, SARS-CoV-2, infects primarily the respiratory tract of humans, causing different symptoms that range from mild-to-moderate (e.g., fever, cough, sore throat, myalgia, fatigue, anosmia, and ageusia) to significant hypoxia due to acute respiratory distress syndrome and severe pneumonia [2]. Various factors increase the severity of the disease, disease rate progression, and mortality rate [3], including demographic factors (e.g., age, gender, pregnancy, post-menopause, poverty, crowding) and certain underlying medical conditions (e.g., high blood pressure, diabetes, obesity). These factors might represent key factors in allowing subsequent viral replication and exacerbating inflammatory and immune-pathological responses [4, 5]. Notably, gastrointestinal symptoms including diarrhea, nausea, and vomiting have also been documented with COVID-19, indicating that the gastrointestinal tract might also be a site of infection [6, 7]. It has been suggested that the presence of gastrointestinal symptoms might be associated with a faster reduction of the viral load through fecal shedding, which was partially suspected in resulting in better outcomes [8].

Similar to many other microbial infections, mild-to-moderate COVID-19 disease occurs in most patients due to the hosts' adequate innate and adaptive immune responses [9]. Given that antiviral immunity is needed to neutralize the virus, inhibit viral replication, and promote the recovery of patients, severe COVID-19 cases may be associated with a dysregulated immune and inflammatory response [10, 11]. High mortality rates were associated with cytokine storms; an excessive production of proinflammatory cytokines that promotes severe acute respiratory distress syndrome and extensive tissue damage, resulting in life-threatening conditions [12]. Furthermore, previous studies suggested that the nutrition status played a pivotal role in the progression of COVID-19 disease [13, 14]. A hypothesis suggested that hyperinflammation and cytokine storms observed in severe COVID-19 cases are associated with an increase in nutrition acquisition, which may contribute to lipotoxicity and damage in non-adipose tissues, particularly in obese patients or individuals with metabolic syndromes [15].

Several mitigation measures have been adopted to prevent the transmission of SARS-CoV-2 and reduce its impact on communities [16]. Non-pharmaceutical prophylactic approaches (such as mask-wearing, washing hands, and physical distancing measures) and COVID-19 vaccines might reduce the amount of virus circulating in and between individuals [17–19]. However, decreasing viral loads might not affect disease severity in secondary cases [20].

Therefore, despite the recent development of effective vaccines against SARS-CoV-2, COVID-19 continues to be problematic, and the probability of exposure to the virus and contracting the disease remains significant across the globe. Notably, vaccinated individuals could potentially still get COVID-19 and transmit live viruses from the upper respiratory tract to others [21]. New variants have rapidly emerged and become dominant worldwide, i.e., the B.1.617.2 (Delta) variant causes more infections and spreads faster than earlier forms of the SARS-CoV-2 [22]. These new variants might also be able to escape from vaccine-induced immunity and other treatment strategies (such as Bamlanivimab/etesevimab). Furthermore, recent data highlighted the strong selective pressure imposed by convalescent plasma therapy, potentially leading to the emergence of SARS-CoV-2 variants with a reduced susceptibility to neutralizing antibodies in immunosuppressed individuals [23]. Indeed, many aspects of the biology of the virus, its evolution and the epidemiology and ramifications of the disease remain unknown and unpredictable. This is very relevant when evaluating the evolution of the disease in low-resource countries where vaccination rates are predicted to be slow. Taken together, these concerns highlight the need to further investigate COVID-19 symptoms, outcomes, exposure, and severity, especially in developed countries with challenges in medical capacity and pandemic preparedness.

It has been widely suggested that after infection with SARS-CoV-2, patients may sometimes develop gastrointestinal symptoms [24]. This is not surprising, given that SARS-CoV-2 interacts with angiotensin-converting enzyme 2 (ACE2) receptor that is expressed on the surface of the epithelial cells that line different organs, including lungs, kidneys, heart, but also the small intestine. Furthermore, it is predicted that altered gut microbiota and associated leaky gut might contribute to severe COVID-19 illness [5]. After all, the gut microbiota provides a wide-ranging essential health benefits and protection to the host [25]. For example, the gut microbiota is known to significantly regulate immune homeostasis through modulating the development and function of the innate and adaptive immune system [26]. Recent studies on gut microbiota suggested that an alteration in the intestinal microbiota composition (i.e., dysbiosis) contributes to various immune-mediated inflammatory diseases [27]. Subsequently, in COVID-19 infections, gut microbiota dysbiosis might play a critical role in determining the clinical outcome of cases with underlying comorbid conditions such as diabetes, hypertension, and obesity [28]. For instance, gut microbiota diversity is generally decreased among elders, and COVID-19 has been more severe and fatal in this susceptible population which highlights a potential role of the gut microbiota in this disease [29]. In this respect, it was suggested that COVID-19 patients are depleted of gut bacteria with known immunomodulatory potential [30]. A recent report showed that poor gut health might adversely affect COVID-19 prognosis by enabling SARS-CoV-2 to leak into the circulatory system and access the surface of the digestive tract and internal organs [5]. Additionally, the inflammation induced by gut dysbiosis represents an important mediator in cardiometabolic and diabetic pathogenesis and could contribute to aggravated COVID-19 in the most vulnerable patients [31]. Given that diet plays a critical role in modulating the gut microbiota, there has been a serious interest in evaluating the health benefits and disease-preventing properties of diet and dietary habits and their association with a favorable patient outcome [32, 33].

In the global effort to control COVID-19, exploiting the indicators of gut health to improve disease outcomes might pose a significant advantage. However, COVID-19 patients can be highly contagious, and the investigations of gut microbiota in these patients have been secondary, understandably due to urgency in treating patients and the many challenges of combating a novel virus or disease in a pandemic. Therefore, investigations into COVID-19 and gut health are scarce and have not fully analyzed possible associations between gut health and disease pathophysiology. Consequently, the current study particularly aimed to demonstrate a

potential association between poor gut health and COVID-19 severe symptoms in the Lebanese community. It should be noted that in parallel to the emergence and spread of the COVID-19, Lebanon has been facing tremendous economic and medical crises, including shortage in healthcare workers, medical supplies, and equipment, together with vaccines in all hospitals across the country. Importantly, Lebanon is currently hosting ~1.5 million Syrian displaced people and ~0.5 million Palestinian refugees, while the total number of poor among the Lebanese population is currently estimated to be ~2.7 million. Given that these populations are known to be susceptible to COVID-19 and the situation of great concern in Lebanon [34, 35], it appears thus important to address COVID-19 in this developing country with severe challenges.

## Material and methods

### 1. Ethics statement

This investigation received the approval of the ethical committee of the Doctoral School of Science and Technology/Lebanese University (CE-EDST-3-2021) in agreement with Lebanese legislation. This article does not involve human samples. Oral informed consent was obtained from the participants before beginning the phone interview. This study was conducted in accordance with the Code of Ethics of the World Medical Association (Declaration of Helsinki). All data were analyzed anonymously.

### 2. Study population

The access to the list of individuals infected by SARS-COV-2 between March and October 2020 in the North Governorate of Lebanon was granted by the Governor's Office. The list is a compilation of information on people who have tested positive for the SARS-COV-2 using a standard q-PCR test that was performed as part of national screening efforts in the healthcare centers certified by the Ministry of Public Health in Lebanon. After removing missing and duplicated data, the total number of individuals was 843. This constituted the final list which is considered as the sole, most valid, and reliable sample available.

### 3. Sample size and responses

The effect of gut health on the severity of COVID-19 symptoms was evaluated by interviewing the individuals in the final list. For this purpose, a questionnaire was programmed into KoBoToolbox software for data collection. Data collectors (n = 12) were trained on conducting phone interviews. An oral consent from each participant was secured using a standard script. To further facilitate and streamline the interviews, the duration of each interview was 10 minutes on average. A representative sample was calculated using the Raosoft online calculator (http://www.raosoft.com/samplesize.html) with a 5% margin of error, a 95% confidence interval, and a 50% response distribution to get the maximum sample size possible. Subsequently, the final sample size of listed individuals that were contacted for interviews was determined to be 500.

However, out of the 500 individuals contacted, a total of 255 interviews were completed. At the end of interviews, participants were asked if they would like to enter a draw to win shopping coupons from a local grocery shop as a compensation for their time and sharing their experiences.

### 4. Variables of interest

COVID-19 symptom severity level was measured on an ordinal scale of 3-point as reported by the participants and ranged from "no symptoms" to "moderate symptoms" and "severe

**Table 1. Gut health scoring.**

| Gut health score[i] | Gut health scoring items | | | | |
|---|---|---|---|---|---|
| | Unhealthy food intake[ii] | Healthy food Intake[iii] | Overweight/obese | Chronic medication | Gut problems[iv] |
| 5 | - | + | - | - | - |
| 4 | - | - | - | - | - |
| | + | -/+ | - | - | - |
| 3 | - | -/+ | Overweight/obese (+) and/or chronic medication (+) | | - |
| 2 | + | -/+ | Overweight/obese (+) and/or chronic medication (+) | | - |
| 1 | -/+ | -/+ | -/+ | -/+ | + |

(-) refers to absence of the corresponding item.

(+) refers to the presence of the corresponding item.

[i] A higher score indicates a better overall gut health.

[ii] Represented by the excessive intake of either fast-food or sugar.

[iii] Represented by the regular intake of at least 2 of the following: whole grains, fermented food, fruits/vegetables, and probiotics.

[iv] Including irritable bowel syndrome (IBS), ulcer, chronic constipation or diarrhea, and/or any gastrointestinal disorder.

symptoms". Sociodemographic variables were also measured. Furthermore, in the analysis, respiratory immunity levels were estimated by the reported number of respiratory infections contracted per year.

For gut health assessment, a composite variable was created based on conceptual reasoning. The latter and scoring of this variable are presented in **Table 1**. Specifically, a comprehensive score was created to estimate the gut health status based on the combination of five complementary parameters that reflected the quality of food intake, the presence of overweight and obesity, the use of chronic medications, and the occurrence of gut disease (**Table 1**). The health score ranged from 1 to 5 (1 = unhealthy; 5 = most healthy). To have the highest score, individuals must: (i) eat healthy food (regular intake of at least 2 of the following: whole grains, fermented food, fruits/vegetables, and probiotic/prebiotic supplements); (ii) not eat unhealthy food (excessive intake of either fast-food or sugar); (iii) not suffer from overweight or obesity; (iv) not use chronic medications; and (v) do not have a gut problem such as irritable bowel syndrome (IBS), ulcer, chronic constipation, and/or any gastrointestinal disorder. Furthermore, patients suffering from a gut problem will directly receive the worst health score, regardless of other variables (**Table 1**).

## 5. Statistical analysis

Data analyses were conducted using the statistical software package Stata/SE version 13 (Stata-Corp, College Station, Texas 77845 USA). Descriptive statistics of the study population and their distribution based COVID-19 symptom severity were performed and presented as mean ± standard deviation (SD) and proportions for continuous and categorical variables, respectively. At the bivariate level, Kendall's tau-b was used to measure and test for the ordinal association of COVID-19 symptom severity with sociodemographic, gut health score, participant's medical condition, dietary and lifestyle, as well as post-COVID-19 infection-related variables such as residual symptoms and precautionary measures. Simple and multiple ordinal logistic regressions were conducted to examine the determinants of COVID-19 symptom severity. Additionally, simple and multiple logistic regressions were conducted to test for the determinants of experiencing any COVID-19 symptoms versus none. Results from regression models were expressed as odds ratio (OR) with 95% confidence intervals (CI). P-values lower than 0.05 were considered statistically significant.

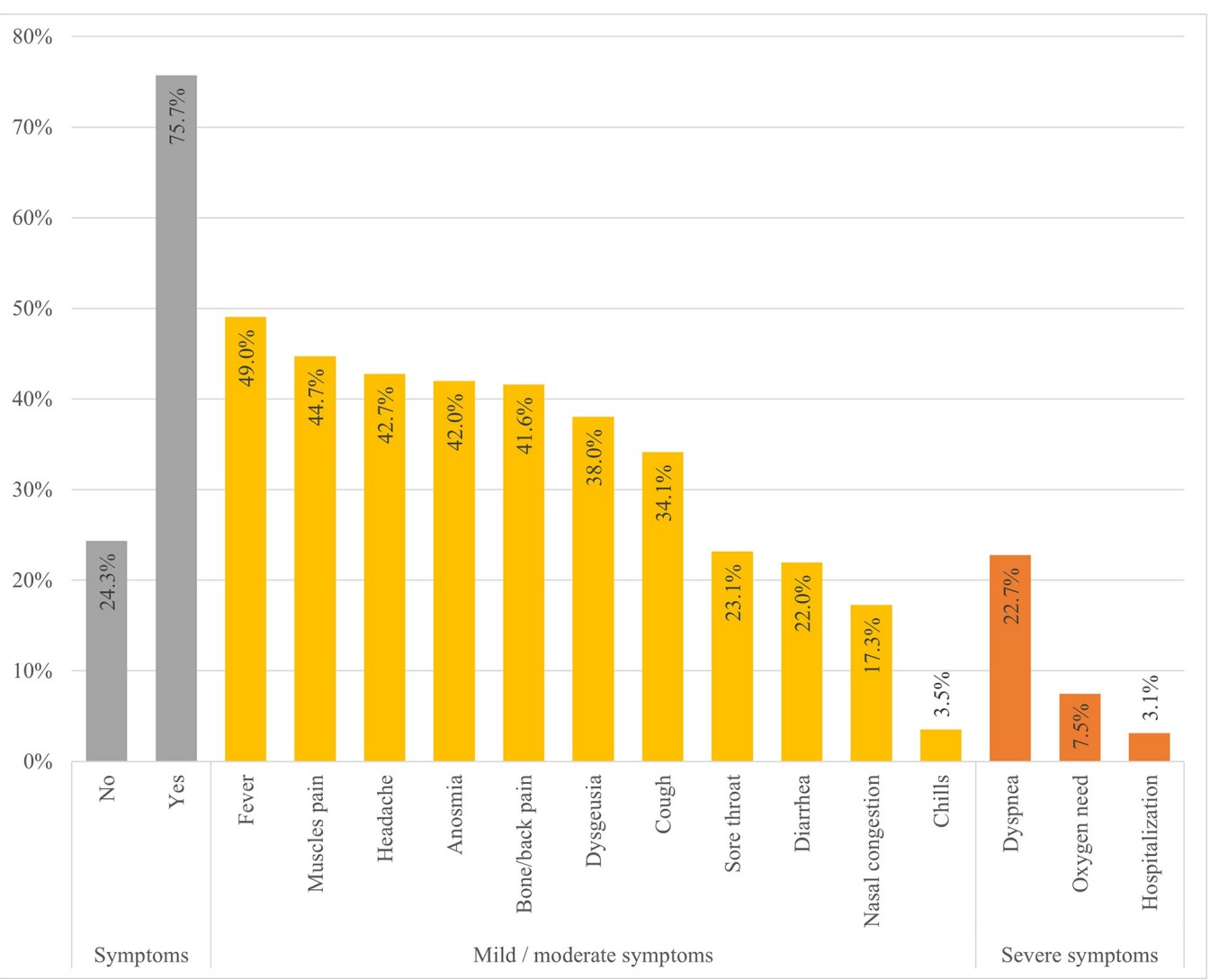

**Fig 1. Clinical characteristics of the COVID-19 patients (n = 255) with telephone assessment.** Symptomatic patients (n = 193) present at least one symptom related to COVID-19.

## Results

Analysis of the data showed that 1 in 4 participants were asymptomatic, while an equal proportion experienced severe symptoms, including dyspnea (22.7%), oxygen need (7.5%), and hospitalization (3.1%). Mild to moderate symptoms were experienced by 51.4%. The overall distribution of symptoms and their classification is summarized in **Fig 1**. The most-reported mild to moderate symptoms were fever (49%), muscle pain (44.7%), headache (42.7%), anosmia (42%), bone/back pain (41.6%), dysgeusia (38%), and cough (34.1%). Less common symptoms included throat pain (23.1%), diarrhea (22%), and nasal congestion (17.3%). The **S1 Table** shows a detailed description of the medical history information of the studied population.

The distribution of sociodemographic variables and their bivariate association with COVID-19 symptoms (treated as an ordinal variable) are shown in **Table 2**. The mean age of the participants was 38.3 ± 0.8 years, and the majority were males (63.9%), married (68.2%),

**Table 2. Description of sociodemographic variables and their association with COVID-19 symptoms.**

| | Total | | No symptoms (n = 62) | | Mild/Moderate (n = 131) | | Severe (n = 62) | | Kendall's tau-b | p-value |
|---|---|---|---|---|---|---|---|---|---|---|
| | n | % | n | % | n | % | n | % | | |
| **Age** (mean ± SD) | 38.3 ± 0.8 | | 40.1 ± 2.2 | | 37.8 ± 1.2 | | 37.8 ± 1.5 | | / | 0.354 |
| **Gender** | | | | | | | | | -0.022 | 0.710 |
| Male | 163 | 63.9 | 38 | 23.3 | 85 | 52.1 | 40 | 24.5 | | |
| Female | 92 | 36.1 | 24 | 26.1 | 46 | 50.0 | 22 | 23.9 | | |
| **Marital status** | | | | | | | | | 0.082 | 0.169 |
| Single | 75 | 29.4 | 19 | 25.3 | 44 | 58.7 | 12 | 16.0 | | |
| Married/divorced | 180 | 70.6 | 43 | 23.9 | 87 | 48.3 | 50 | 27.8 | | |
| **Working status** | | | | | | | | | 0.147 | **0.014** |
| No | 85 | 33.3 | 27 | 31.8 | 44 | 51.8 | 14 | 16.5 | | |
| Yes | 170 | 66.7 | 35 | 20.6 | 87 | 51.2 | 48 | 28.2 | | |
| **Socioeconomic status (SES)** | | | | | | | | | -0.025 | 0.661 |
| Below poverty line/Low SES | 89 | 34.9 | 26 | 29.2 | 35 | 39.3 | 28 | 31.5 | | |
| Middle SES | 144 | 56.5 | 31 | 21.5 | 83 | 57.6 | 30 | 20.8 | | |
| High SES | 22 | 8.6 | 5 | 22.7 | 13 | 59.1 | 4 | 18.2 | | |

Data are presented as mean ± SD for the continuous variable age and as frequency and percentage for categorical variables.

currently employed (66.7%), and of a middle socioeconomic status (SES) (56.5%). Notably, a significant association was found between COVID-19 symptoms and working status, where employed participants were more likely to experience symptoms (Kendall's tau-b = 0.147, p-value 0.014). No linear trend was observed with age, gender, marital status, and SES at the bivariate level.

Gut health main elements and gut health scores and their association with COVID-19 symptoms were described in **Table 3**. Healthy food intake was reported by 78% of the participants, while unhealthy food intake was reported by half of the participants. About two-thirds (62%) of the participants were overweight or obese, and the majority do not take any chronic medications (71%). Moreover, none reported the use of antibiotics. Gut problems were reported by 22.7% of the study participants. Each gut health score item independently was not significantly linked to COVID-19 symptoms except for excessive sugar or fast-food intake; however, participants who reported unhealthy food intake were more likely to experience severe symptoms (Kendall's tau-b = 0.118, p-value 0.049). When all items were taken into consideration, a negative correlation was observed between gut health score and COVID-19 symptoms, where lowest gut health scores were associated with severe COVID-19 symptoms (Kendall's tau-b = -0.153, p-value 0.004).

**Table 4** summarized the description of other variables including the participants' medical history, other dietary and lifestyle variables and post-COVID-19 infection variables, and their association with COVID-19 symptoms. Most of the participants (85.9%) had a positive blood rhesus factor and had no history of chronic diseases (80%). Neither rhesus factor nor chronic disease history had an association with COVID-19 severity at the bivariate level. About half of the study participants had 0 or 1 episode of flu-like symptoms per year, and 27% and 21.2% experienced 2 and 3 or more episodes of flu-like symptoms per year, respectively. The bivariate analysis showed that the higher the number of flu-like illness episodes the worse the COVID-19 symptoms (Kendall's tau-b = 0.194, p-value 0.001). Thirty-four percent (45/132) of those who reported a flu-like illness count of ≤1 per year did not experience any symptoms compared to 11.6% (8/69) and 16.7% (9/54) for those reporting 2 and 3 or more flu-like illness

**Table 3. Description of gut health core elements and gut health score and their association with COVID-19 symptoms.**

| | Total | | No symptoms (n = 62) | | Mild/Moderate (n = 131) | | Severe (n = 62) | | Kendall's tau-b | p-value |
|---|---|---|---|---|---|---|---|---|---|---|
| | n | % | n | % | n | % | n | % | | |
| *Gut health score elements* | | | | | | | | | | |
| **Unhealthy food intake** | | | | | | | | | 0.118 | **0.049** |
| No | 126 | 49.4 | 35 | 27.8 | 67 | 53.2 | 24 | 19.0 | | |
| Yes | 129 | 50.6 | 27 | 20.9 | 64 | 49.6 | 38 | 29.5 | | |
| **Healthy food intake** | | | | | | | | | 0.065 | 0.280 |
| No | 56 | 22.0 | 17 | 30.4 | 27 | 48.2 | 12 | 21.4 | | |
| Yes | 199 | 78.0 | 45 | 22.6 | 104 | 52.3 | 50 | 25.1 | | |
| **Body mass index (BMI)[i]** | | | | | | | | | 0.103 | 0.086 |
| Underweight/normal | 93 | 37.1 | 25 | 26.9 | 51 | 54.8 | 17 | 18.3 | | |
| Overweight/obese | 158 | 62.9 | 34 | 21.5 | 79 | 50.0 | 45 | 28.5 | | |
| **Chronic medication** | | | | | | | | | 0.000 | 1.000 |
| No | 181 | 71.0 | 45 | 24.9 | 91 | 50.3 | 45 | 24.9 | | |
| Yes | 74 | 29.0 | 17 | 23.0 | 40 | 54.1 | 17 | 23.0 | | |
| **Gut problems** | | | | | | | | | 0.102 | 0.087 |
| Absence | 197 | 77.3 | 52 | 26.4 | 101 | 51.3 | 44 | 22.3 | | |
| Presence | 58 | 22.7 | 10 | 17.2 | 30 | 51.7 | 18 | 31.0 | | |
| *Gut health score* | | | | | | | | | | |
| **Gut health score[ii]** | | | | | | | | | -0.153 | **0.004** |
| 1 Gut problems | 58 | 22.8 | 10 | 17.2 | 30 | 51.7 | 18 | 31.0 | | |
| 2 | 68 | 26.8 | 13 | 19.1 | 31 | 45.6 | 24 | 35.3 | | |
| 3 | 61 | 24.0 | 18 | 29.5 | 33 | 54.1 | 10 | 16.4 | | |
| 4 | 41 | 16.1 | 13 | 31.7 | 23 | 56.1 | 5 | 12.2 | | |
| 5 | 26 | 10.2 | 7 | 26.9 | 14 | 53.8 | 5 | 19.2 | | |

[i] BMI was stratified according to the WHO criteria (World Health Organization. Body mass index—BMI. Available online: http://www.euro.who.int/en/health-topics/disease-prevention/nutrition/a-healthy-lifestyle/body-mass-index-bmi (accessed on 10 April 2021).

[ii] A higher score indicates a better overall gut health.

count per year, respectively. Also, vomiting and diarrhea count were recorded but no linear trend was shown with COVID-19 symptoms. About two-thirds (68.6%) of the study participants reported regularly drinking tea or coffee and a negative correlation was detected with COVID-19 symptom at the crude level. The proportion of participants who did not experience any symptoms was higher among coffee or tea drinkers (28.6% vs 15% for non-tea or coffee drinkers). Furthermore, coffee or tea drinkers experienced less severe symptoms (21.7% vs 30% for non-tea or coffee drinkers). About 1 in 4 (26.9%) of participants who experienced any COVID-19 symptoms reported having residual symptoms after recovery including fatigue (44.2%), anosmia (21.2%), and cough (17.3%) (**S2 Table**). No association between residual symptoms and COVID-19 severity was observed at the bivariate level. Although only 16.7% of participants reported not taking precautionary measures post-COVID-19 infection, a positive correlation was found between this variable and COVID-19 symptoms, where the higher the level of COVID-19 symptoms experienced the less likely it is that the participant was taking precautionary measures post-recovery (**Table 4**).

The findings of the ordinal logistic regression to identify the determinants of COVID-19 symptoms (ordinal dependent variable) are summarized in **Table 5**. Model 1 is unadjusted, while in model 2 each independent variable was entered separately together with age and

**Table 4. Description of other variables and their association with COVID-19 symptoms.**

| | Total | | No symptoms (n = 62) | | Mild/Moderate (n = 131) | | Severe (n = 62) | | Kendall's tau-b | p-value |
|---|---|---|---|---|---|---|---|---|---|---|
| | n | % | n | % | n | % | n | % | | |
| *Patients' medical history* | | | | | | | | | | |
| **Rhesus factor** | | | | | | | | | 0.077 | 0.213 |
| ABO- | 18 | 7.1 | 6 | 33.3 | 9 | 50.0 | 3 | 16.7 | | |
| ABO+ | 219 | 85.9 | 47 | 21.5 | 115 | 52.5 | 57 | 26.0 | | |
| **Chronic diseases** | | | | | | | | | 0.042 | 0.117 |
| Absence | 204 | 80.0 | 52 | 25.5 | 107 | 52.5 | 45 | 22.1 | | |
| Presence | 51 | 20.0 | 10 | 19.6 | 24 | 47.1 | 17 | 33.3 | | |
| **Flu-like symptoms (count per year)** | | | | | | | | | 0.194 | **0.001** |
| 0–1 | 132 | 51.8 | 45 | 34.1 | 64 | 48.5 | 23 | 17.4 | | |
| 2 | 69 | 27.1 | 8 | 11.6 | 37 | 53.6 | 24 | 34.8 | | |
| 3 or more | 54 | 21.2 | 9 | 16.7 | 30 | 55.6 | 15 | 27.8 | | |
| **Vomiting and/or diarrhea (count per year)** | | | | | | | | | 0.052 | 0.371 |
| 0–1 | 196 | 76.9 | 47 | 24.0 | 106 | 54.1 | 43 | 21.9 | | |
| 2 | 33 | 12.9 | 10 | 30.3 | 12 | 36.4 | 11 | 33.3 | | |
| 3 or more | 26 | 10.2 | 5 | 19.2 | 13 | 50.0 | 8 | 30.8 | | |
| *Dietary and lifestyle* | | | | | | | | | | |
| **Smoking** | | | | | | | | | -0.118 | 0.470 |
| No | 145 | 56.9 | 28 | 19.3 | 85 | 58.6 | 32 | 22.1 | | |
| Yes | 110 | 43.1 | 34 | 30.9 | 46 | 41.8 | 30 | 27.3 | | |
| **Excessive coffee/tea consumption** | | | | | | | | | -0.138 | **0.021** |
| No | 80 | 31.4 | 12 | 15.0 | 44 | 55.0 | 24 | 30.0 | | |
| Yes | 175 | 68.6 | 50 | 28.6 | 87 | 49.7 | 38 | 21.7 | | |
| *Post-COVID-19 infection* | | | | | | | | | | |
| **Residual symptoms** | | | | | | | | | 0.032 | 0.655 |
| No | 141 | 73.1 | - | - | 97 | 68.8 | 44 | 31.2 | | |
| Yes | 52 | 26.9 | - | - | 34 | 65.4 | 18 | 34.6 | | |
| **Precautionary measures** | | | | | | | | | -0.142 | **0.020** |
| No | 41 | 16.7 | 4 | 9.8 | 22 | 53.7 | 15 | 36.6 | | |
| Yes | 204 | 83.3 | 48 | 23.5 | 109 | 53.4 | 47 | 23.0 | | |

gender, and in model 3 the following variables were entered in the model: age, gender, chronic disease, working status, flu-like illness counts, smoking, coffee/tea consumption, and gut health. Confirming the bivariate associations in **Tables 1–3**, a significant association was detected between COVID-19 symptoms and each of the following variables: working status, flu-like illness counts, coffee/tea consumption, and gut health score. After adjusting for age and gender in model 2, the association between COVID-19 symptoms and chronic diseases reached statistical significance (OR: 1.93, 95%CI: 1.03–3.62). However, after adjusting for further potential confounders in model 3, the association was lost. In model 3, being employed persisted as a risk factor for worse COVID-19 symptoms (OR: 2.12, 95%CI: 1.17–3.85) as well as the higher flu-like illness count per year (OR of having 2 vs 0 or 1 flu-like illness count: OR: 3.04, 95%CI: 1.69–5.45; of having 3 or more vs 0 or 1 flu-like illness count: OR: 2.08, 95%CI: 1.1–3.94). However, coffee or tea consumption association with COVID-19 symptoms was lost after adjustments in model 3. Regarding gut health, after the adjustment for potential confounders in model 3, the statistical analysis showed that for every level increase in gut health (better overall gut health) the chances of having worse COVID-19 symptoms (that is having

**Table 5. Determinants of COVID-19 symptoms using simple and multiple ordinal logistic regression.**

| | Model 1[i] | | | Model 2[ii] | | | Model 3[iii] | | |
|---|---|---|---|---|---|---|---|---|---|
| | OR | p-value | 95% CI | adj. OR | p-value | 95% CI | adj. OR | p-value | 95% CI |
| *Sociodemographic variables* | | | | | | | | | |
| Age | 0.99 | 0.354 | (0.98–1.01) | 0.99 | 0.337 | (0.98–1.01) | 0.98 | 0.051 | (0.96–1) |
| **Gender** (females vs males) | 0.91 | 0.708 | (0.56–1.48) | 0.89 | 0.654 | (0.55–1.46) | 1.27 | 0.434 | (0.7–2.28) |
| **Marital status** (married/divorced vs single) | 1.42 | 0.174 | (0.86–2.35) | **1.97** | **0.047** | **(0.96–1.00)** | 1.64 | 0.130 | (0.86–3.11) |
| **Working** (employed vs unemployed) | **1.88** | **0.014** | **(1.14–3.09)** | **2.13** | **0.010** | **(1.2–3.81)** | **2.00** | **0.023** | **(1.1–3.65)** |
| **Socioeconomic status (SES)** | | | | | | | | | |
| Middle SES (vs low SES) | 0.92 | 0.746 | (0.55–1.53) | 0.88 | 0.640 | (0.53–1.48) | | | |
| High SES (vs low SES) | 0.83 | 0.680 | (0.35–2) | 0.84 | 0.690 | (0.35–2) | | | |
| *Patients' medical condition* | | | | | | | | | |
| **Rhesus factor** (ABO+ vs ABO-) | 1.80 | 0.209 | (0.72–4.50) | 1.78 | 0.220 | (0.71–4.45) | | | |
| **Chronic disease** (presence vs absence) | 1.61 | 0.113 | (0.89–2.91) | **1.93** | **0.040** | **(1.03–3.62)** | 1.60 | 0.155 | (0.84–3.05) |
| **Flu-like illness** (count per year) | | | | | | | | | |
| 2 (vs 0 or 1) | **3.06** | **<0.001** | **(1.73–5.42)** | **3.04** | **<0.001** | **(1.72–5.39)** | **2.85** | **0.001** | **(1.58–5.15)** |
| 3 or more (vs 0 or 1) | **2.20** | **0.012** | **(1.19–4.05)** | **2.12** | **0.017** | **(1.14–3.95)** | **2.04** | **0.028** | **(1.08–3.86)** |
| **Vomiting and/or diarrhea** (count per year) | | | | | | | | | |
| 2 (vs 0 or 1) | 1.16 | 0.695 | (0.56–2.4) | 1.12 | 0.760 | (0.54–2.34) | | | |
| 3 or more (vs 0 or 1) | 1.45 | 0.350 | (0.67–3.15) | 1.40 | 0.395 | (0.64–3.07) | | | |
| *Dietary and lifestyle* | | | | | | | | | |
| **Smoking** | 0.84 | 0.464 | (0.52–1.35) | 0.82 | 0.425 | (0.51–1.33) | 0.70 | 0.171 | (0.42–1.17) |
| **Coffee/tea consumption** | **0.55** | **0.021** | **(0.33–0.92)** | **0.55** | **0.022** | **(0.33–0.92)** | 0.67 | 0.115 | (0.41–1.1) |
| *Gut health score* | | | | | | | | | |
| **Patients' gut health** | **1.30** | **0.007** | **(1.07–1.56)** | **1.35** | **0.003** | **(1.11–1.63)** | **1.31** | **0.008** | **(1.07–1.61)** |

Abbreviations: OR = odds ratio; CI = confidence interval, adj = adjusted.

[i] Model 1 is unadjusted.

[ii] In model 2 each independent variable was entered separately together with age and gender.

[iii] In model 3 the following variables were entered in the model: age, gender, chronic disease, working status, flu-like illness count, smoking, coffee/tea consumption, and gut health.

mild/moderate or severe symptoms vs no symptoms or having severe symptoms vs either no symptoms or mild/moderate symptoms) decreases by 33%.

Forty-three percent of the study participants were smokers, and no linear trend was detected between smoking and COVID-19 symptoms. However, when comparing participants who experienced any symptoms (mild/moderate or severe combined) vs those who did not experience any, the latter group was more likely to be smokers (and this association remained even after adjusting for potential confounders) (**Table 6**). Interestingly, in contrary to the comparison between the three levels of COVID-19 symptoms (absence, mild-to-moderate, and severe symptoms), coffee or tea consumption association with COVID-19 symptoms has remained significant after the comparison of COVID-19 symptoms (absence vs presence) using simple and multiple logistic regression analyses (**Table 6**).

## Discussion

In this cross-sectional survey, around 24.3% of confirmed COVID-19 patients were asymptomatic which is higher than the results obtained in a previous systematic review which

**Table 6. Determinants of having any COVID-19 symptoms (vs none) using simple and multiple logistic regression.**

| | Model 1[i] | | | Model 2[ii] | | | Model 3[iii] | | |
|---|---|---|---|---|---|---|---|---|---|
| | OR | p-value | 95% CI | adj OR | p-value | 95% CI | adj OR | p-value | 95% CI |
| *Sociodemographic variables* | | | | | | | | | |
| Age | 0.99 | 0.267 | (0.97–1.01) | 0.99 | 0.245 | (0.97–1.01) | 0.99 | 0.244 | (0.97–1.01) |
| **Gender** (females vs males) | 0.86 | 0.620 | (0.48–1.55) | 0.83 | 0.544 | (0.46–1.51) | 0.81 | 0.524 | (0.42–1.55) |
| **Marital status** (married/divorced vs single) | 1.14 | 0.682 | (0.62–2.09) | 1.48 | 0.282 | (0.73–3.01) | | | |
| **Working status** (employed vs unemployed) | 1.80 | 0.051 | (1–3.24) | 1.97 | 0.060 | (0.97–3.99) | | | |
| **Socioeconomic status (SES)** | | | | | | | | | |
| Middle SES (vs below poverty line / low SES) | 1.50 | 0.186 | (0.82–2.76) | 1.44 | 0.241 | (0.78–2.66) | | | |
| High SES (vs low SES) | 1.40 | 0.545 | (0.47–4.2) | 1.41 | 0.539 | (0.47–4.24) | | | |
| *Patients' medical condition* | | | | | | | | | |
| **Rhesus factor** (ABO+ vs ABO-) | 1.83 | 0.251 | (0.65–5.13) | 1.81 | 0.261 | (0.64–5.12) | | | |
| **Chronic disease** (presence vs absence) | 1.40 | 0.383 | (0.66–3) | 1.76 | 0.171 | (0.78–3.97) | | | |
| **Flu-like illness** (count per year) | | | | | | | | | |
| 2 (vs 0 or 1) | **3.94** | **0.001** | **(1.74–8.96)** | **3.88** | **0.001** | **(1.71–8.84)** | **4.04** | **0.001** | **(1.73–9.43)** |
| 3 or more (vs 0 or 1) | **2.59** | **0.020** | **(1.16–5.76)** | **2.45** | **0.031** | **(1.09–5.54)** | **2.58** | **0.029** | **(1.1–6.03)** |
| **Vomiting and/or diarrhea** (count per year) | | | | | | | | | |
| 2 (vs 0 or 1) | 0.73 | 0.438 | (0.32–1.63) | 0.69 | 0.369 | (0.3–1.56) | | | |
| 3 or more (vs 0 or 1) | 1.32 | 0.592 | (0.47–3.71) | 1.28 | 0.639 | (0.45–3.61) | | | |
| *Dietary and lifestyle* | | | | | | | | | |
| **Smoking** (Yes vs No) | **0.53** | **0.034** | **(0.3–0.95)** | **0.51** | **0.023** | **(0.28–0.91)** | **0.44** | **0.013** | **(0.23–0.84)** |
| **Excessive coffee/tea consumption** (Yes vs No) | **0.40** | **0.004** | **(0.22–0.75)** | **0.41** | **0.005** | **(0.22–0.76)** | **0.49** | **0.034** | **(0.25–0.95)** |
| *Gut health score* | **1.23** | **0.069** | **(0.98–1.53)** | **1.28** | **0.038** | **(1.01–1.64)** | **1.33** | **0.029** | **(1.03–1.72)** |

Abbreviations: OR = odds ratio; CI = confidence interval; adj = adjusted.

[i] Model 1 is unadjusted.

[ii] In model 2 each independent variable was entered separately together with age and gender.

[iii] In model 3 the following variables were entered in the model: age, gender, flu-like illness count, smoking, coffee/tea consumption, and gut health.

showed that 15.6% (95% CI, 10.1%-23.0%) of cases presented an asymptomatic infection [36]. This could be attributed to the fact that the participants in this study were mainly young adults (the mean age of the population is 38.3, with an age range from 16 to 85) who normally express lower physical COVID-19 symptoms. Additionally, COVID-19 related death cases were not included in this study [37]. Regarding symptomatic patients, 68% and 32% showed mild-to-moderate and severe symptoms, respectively. Globally, the most prevalent features in patients with laboratory confirmed COVID-19 was fever; experienced by 49% of patients [37, 38]. While dry cough and diarrhea were less commonly reported, anosmia and dysgeusia seem to be frequent findings among COVID-19 patients in this investigation. The data emphasized that the onset of loss of smell and taste symptoms, particularly in the absence of symptoms indicative of nasal obstruction, is a strong indicator for a diagnosis of COVID-19 [39]. Furthermore, dyspnea was the most prevalent severe symptom observed among the included population. This is consistent with previous reports demonstrating that patients who are hospitalized with severe SARS-CoV-2 infection tend to have shortness of breath [40–42]. In addition to the virulence of SARS-CoV-2 variants, demographic characteristics such as age, gender, and biological differences in factors such as immune regulation may be associated with COVID-19 disease manifestation, susceptibility, and progression [43].

Gut microbiota is crucial to maintain an anti-inflammatory and healthy environment in the digestive tract ecosystem [44]. The microbiota contributes to the development of the immune system, protecting humans from infectious diseases [45]. The alteration in the intestinal microbiota has been associated with several pathologies, such as obesity, inflammatory bowel syndrome, cancer, and autoimmune diseases [46, 47]. There are numerous factors leading to gut dysbiosis such as aging [48], diet [49], chronic medical conditions [50], probiotic supplementation [51], and antimicrobial use [52], which subsequently affects host health. For instance, the gut microbiota of elderly people is rich with pro-inflammatory bacteria at the expense of beneficial microorganisms [48]. Gut dysbiosis could lead to serious chronic inflammation and mucosal tissue damage which increase the risk of developing serious infectious diseases like *Clostridium difficile* infections [53]. Unfortunately, few studies explored the association between the gut microbiota status and COVID-19 severity, clinical complications, and related death. However, available literature predicted that the unhealthy status of gut microbiota might represent an underscored risk factor [5, 30, 54].

Although the determination of gut composition and health status by using biological approaches such as metagenomic analysis is possible, there are many obstacles to reach definitive outcomes. These include the difficulty of taking stool samples from COVID-19 patients who are contagious and therapy which could have effect on patients' microbiota. Therefore, epidemiological studies can be a relevant alternative and can use advanced statistics to provide evidence on potential associations. For this purpose, this study has established a comprehensive gut health score reflecting patients' behavior and gut health status. The first indicator of a potential healthy gut microbiota is the quality of food [55]. Participants were therefore asked about the consumption of healthy and unhealthy food. The non-compliance of regular consumption of dietary fiber has a negative effect on the health status and diversity of gut microbiota [56], and could be associated with the degradation of the colonic mucus barrier on the gut epithelial lining [57]. In contrast, ingestion of dietary fiber sources stimulates microbial proliferation and produces microbially derived end products such as short-chain fatty acids [58]. These compounds support gut health through a number of local effects, including the maintenance of intestinal barrier integrity, production of mucus, protection versus inflammation, and diminishing the risk of colorectal cancer [59]. Moreover, the consumption of either fermented food (e.g., milk, yogurt, pickles, and cheese) or the intake of probiotic and prebiotic supplements have a positive impact on gut health [60]. In contrast, high dietary sugar or processed food were found to decrease gut microbiota diversity [61], stimulate the overgrowth of gut pathobionts and their virulence factors [62, 63], and potentially lead to metabolic disorders [64, 65]. Besides the dietary intake indicator, the gut health score depended on body mass index that predicts overweight and obesity and on the use of chronic medications and the occurrence of gut problems. The gut microbiota of overweight and obese individuals are characterized by specific alterations in the composition and function of the intestinal human microbiota compared to healthy individuals [66]. Furthermore, the chronic use of proton pump inhibitors (PPIs), anti-diabetes, anti-obesity non-steroidal anti-inflammatory, and anti-depression drugs has direct effects on gut microbiota composition and consequently on gut health [67]. Similar observations were reported in the literature among individuals suffering from gut diseases such as IBS, ulcer, chronic constipation, among others [68–70].

To our knowledge, the present study represents the first report describing a significant association between poor gut health score and severe COVID-19 symptoms among Lebanese patients. Although this study did not include COVID-19-related death cases, due to the retrospective design which decreased the percentage of severe cases, COVID-19 severe symptoms were still more common among patients with poor gut health scores. Neither the age nor the gender of the study population was associated with severe COVID-19 symptoms. Only

patients eating unhealthy food or having a low gut health score presented more severe manifestations. Logistic regression models confirmed that poor gut health status was significantly associated with the presence of COVID-19 related symptoms (OR: 1.33, 95%CI: 1.03–1.72; P = 0.029) and their severity (OR: 1.31, 95%CI: 1.07–1.61; P = 0.008). Interestingly, these observations are consistent with previous reports suggesting a putative link between poor gut health and vulnerability to severe COVID-19 illnesses [29, 71–74]. A recent study stated that COVID-19-positive older people presenting with chronic medical conditions such as high blood pressure, diabetes, and obesity are at a higher risk for hospitalization and mortality than older people without these chronic conditions [75]. These latter are associated with numerous changes in the structure of the gut microbiota of humans [76–79]. Interestingly, Gu *et al*. revealed that a significant reduction in bacterial diversity in gut microbiota among COVID-19 patients compared to healthy controls [71]. A depletion in the abundance of beneficial bacteria was reported, particularly those belonging to the class Clostridia [74] and *Ruminococcaceae* and *Lachnospiraceae* families [71]. Furthermore, numerous gut commensals with known immunomodulatory potential including *Bifidobacterium* species but also *Eubacterium rectale* and *Faecalibacterium prausnitzii* were underrepresented in COVID-19 patients compared with non-COVID-19 individuals [30].

This study also compiled other information that could be associated with the severity of COVID-19 illness. Participants who had more than one episode of flu-like illness per year are more likely to present more severe COVID-19 symptoms (OR of having 2 vs 0 or 1 flu-like illness count: OR: 3.04, 95%CI: 1.69–5.45, P = 0.001; of having 3 or more vs 0 or 1 flu-like illness count: OR: 2.08, 95%CI: 1.1–3.94, P = 0.028). Flu-like illness and COVID-19 are both contagious respiratory illnesses, but they are caused by different viruses. The data in this study suggested that becoming ill with other respiratory viruses might worsen the outcome of COVID-19. This could be explained by the fact that viral RNA infections share similar comorbidities or associated with an impaired immune response. Upon infection with RNA respiratory viruses, type 1 immune responses are initiated by sensor respiratory cells such as airway epithelial cells, alveolar macrophages, and dendritic cells. Independently of viral infection, these sensors express pattern recognition receptors (PRRs) that recognize viral pathogens, particularly Toll-like receptor-3 (TLR-3) and RIG-I-like receptors in the case of SARS-CoV-2 infection [80, 81].

This study has a few limitations. Due to challenges of data collection in Lebanon and the enrollment of a limited number of individuals, the epidemiologic significance of these results requires further experimental investigations. However, given that the country is experiencing economic and political crises, there are limited funding opportunities and logistics are difficult. Therefore, we were unable to collect fecal samples and study gut microbiota using metagenomic and culturomic approaches. Integrating epidemiological data with advanced diagnostic methods is essential to better understand the association between gut health status and severity of COVID-19 illness and predict reliable outcomes.

In conclusion, COVID-19 represents an overwhelming challenge in Lebanon and is further aggravated by the current severe medical and economic crises. To our knowledge, this study is the first investigation describing a significant association between poor gut health score and severe COVID-19 symptoms among Lebanese patients. In addition to the importance of the findings at the global level, this study could help local public health workers to promote healthy habits among residents in Lebanon, particularly during the crises. For a better understanding of this association, further cohort studies including more individuals and performing molecular tests are needed. Given that COVID-19 is a global threat that transcends borders, the results in this study call on international stakeholders to play a key role in helping Lebanon to combat the pandemic.

## Supporting information

**S1 Table. Detailed description of medical history information.**
(DOCX)

**S2 Table. Detailed description of healthy/unhealthy food intake and residual symptoms.**
(DOCX)

## Acknowledgments

The authors would like to thank Ms. Israa Khoder, Ms. Souad El Turk, Ms. Rehab El Turk, Ms. Aisha El Sawalhy, Ms. Rim Taleb, Ms. Jinan El Ahmad, Ms. Habiba El Gher, Ms. Najwa Sankari, Mr. Hamed Ahmad, Ms. Sabah Merhabi, Ms. Roukaya El Boustani, and Mr. Mohamad El Halabi for their contribution in collecting data and for conducting the phone-based surveys.

## Author Contributions

**Conceptualization:** Imad Al Kassaa, Sarah El Omari.

**Investigation:** Imad Al Kassaa, Sarah El Omari, Nada Abbas, Nicolas Papon, Djamel Drider, Issmat I. Kassem, Marwan Osman.

**Methodology:** Imad Al Kassaa, Marwan Osman.

**Supervision:** Imad Al Kassaa, Marwan Osman.

**Validation:** Imad Al Kassaa, Issmat I. Kassem, Marwan Osman.

**Visualization:** Imad Al Kassaa, Sarah El Omari, Nada Abbas, Marwan Osman.

**Writing – original draft:** Imad Al Kassaa, Sarah El Omari, Nada Abbas, Issmat I. Kassem, Marwan Osman.

**Writing – review & editing:** Imad Al Kassaa, Nicolas Papon, Djamel Drider, Issmat I. Kassem, Marwan Osman.

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
