## [Decision Letter · Decision Letter 0]

20 Sep 2021

PONE-D-21-25958High association of COVID-19 severity with poor gut health score in Lebanese patientsPLOS ONE

Dear Dr. Osman,

Thank you for submitting your manuscript to PLOS ONE. After careful consideration, we feel that it has merit but does not fully meet PLOS ONE’s publication criteria as it currently stands. Therefore, we invite you to submit a revised version of the manuscript that addresses the points raised during the review process.

We look forward to receiving your revised manuscript.

Kind regards,

Sanjay Kumar Singh Patel, Ph.D.

Academic Editor

PLOS ONE

Journal Requirements:

4. Please ensure that you refer to Figure 1 in your text as, if accepted, production will need this reference to link the reader to the figure.

Reviewers' comments:

Reviewer #1: The manuscript by Kassaa et al. “High association of COVID-19 severity with poor gut health score in Lebanese patients” is interesting. The manuscript requires minor revision before its publication in PLOS One.

Comments

1. The authors avoid the uses of standard deviation in the text and the uses of third person i.e., we etc.

2. Page 3, line 14, the author may add few information about initial prevention approaches (benefits) for COVID-19 i.e. social distancing, improving health (by anti-covid biomolecules), and future challenges by their high mutation ability (variants) and treatments with citations doi: 10.1371/journal.pone.0252300; doi: 10.1007/s12088-020-00893-4; doi: 10.1007/s12088-020-00908-0).

3. Page 4, line 19, please add information about correlation of diet, microbiota and COVID (doi: 10.1007/s12088-020-00908-0). Also, add such information in discussion (minor).

4. The aim and objectives of this study are not clear.

5. Fig. 1 quality may be improved (high resolution).

6. At least one additional Figure (illustration) may be provided as to highlight the summary or prospect of this study.

Reviewer #2: This study investigated the possible associations between patients' gut health and COVID-19 disease pathophysiology. It is found that there is a significant association between poor gut health score and the disease symptom severity among Lebanese COVID-19 patients. The results presented in this study is very timely, comprehensive and pioneering, and would be a big contribution to the fighting against COVID-19 disease if it can be published. This paper would be more compelling if the following minor revision points on gut microbiota and inflammation could be included:

1) In talking about microorganism infection or human microbiota, the human host immunity response has to be taken into consideration. Because of the host immune defences, most of the viral and bacterial infections are self-limiting. There is no difference with the SARS-COV-2 virus infection, that’s why most of the COVID-19 cases are asymptomatic or mild.

2) Human immune system also has pivotal role in nutrition acquisition from the human microbiota during an acute infection, as both the microorganisms and the damaged host tissue become the source of nutrition with the help of the immunity.

3) Although the virulence of the microorganism determines the damages made by the microorganisms to the host cells before they are cleared by host immunity, this microbial or viral virulence alone doesn’t account for the virulence of the disease. The most part of the virulence of an infectious disease is actually the result of the overactive inflammation response of our immune system.

4) As a protective response of the human body to remove injurious stimuli and initiate tissue repairing/regeneration process, inflammation is the physiological machinery of immune system to tissue damage. Yet, lipotoxicity as a result of overnutrition will prevent the tissue healing process from happening. Moreover, in the state of over-nutrition, lipotoxicity and tissue damage will form a malignant positive feedback loop, which escalates the inflammation situation and accounts for most of the severity of the COVID-19 disease. So in order to attenuate hyperinflammation, restrictive eating should be performed during infection so that the transient overnutrition from acute infection can be dissipated.

5) Illness-induced gastrointestinal symptoms like loss of appetite (illness induced anorexia), diarrhea (cause by certain form of dysbiosis), nausea, vomiting and abdominal pain might be our body’s normal physiological reaction to infection induced over-nutrition, in order to dissipate the nutrition surge of an acute infection. So if these symptoms do not lead to complications like dehydration, they may be beneficial to the host in fighting COVID-19.

There are two typos in the manuscript: on page 8, line 5: “Error! Reference source not found.” should be “Figure 1”; on page 11, line 10, “dry was less commonly reported” should be “diarrhoea was less commonly reported”. The following related references may be included in this review to provide a more complete picture on microbiome, infection, immunity and inflammation:

Reference:

Azkur AK, Akdis M, Azkur D, Sokolowska M, van de Veen W, Bruggen MC, O'Mahony L, Gao YD, Nadeau K, Akdis CA (2020) Immune response to SARS-CoV-2 and mechanisms of immunopathological changes in COVID-19. Allergy, 75(7):1564-1581. DOI: 10.1111/all.14364

Yu BX, Yu LG & Klionsky DJ. Nutrition Acquisition by Human Immunity, Transient Overnutrition and the Cytokine Storm in Severe Cases of COVID-19. Medical Hypotheses, 155, 110668 (2021). DOI: 10.1016/j.mehy.2021.110668.

Papa A, Covino M, Pizzolante F, Miele L, Lopetuso LR, Bove V, Iorio R, Simeoni B, Vetrone LM, Tricoli L, Mignini I, Schepis T, D'Alessandro A, Coppola G, Nicoletti T, Visconti E, Rapaccini G (2020) Gastrointestinal symptoms and digestive comorbidities in an Italian cohort of patients with COVID-19. Eur Rev Med Pharmacol Sci 24:7506-7511. DOI: 10.26355/eurrev_202007_21923

Troisi J, Venutolo G, Pujolassos Tanyà M, Delli Carri M, Landolfi A, Fasano A. COVID-19 and the gastrointestinal tract: Source of infection or merely a target of the inflammatory process following SARS-CoV-2 infection? World J Gastroenterol 2021; 27(14): 1406-1418. DOI: 10.3748/wjg.v27.i14.1406

---

## [Author Response · Author response to Decision Letter 0]

2 Oct 2021

Please find a revised version of our manuscript. As requested, we answered all reviewers' comments and suggestions. All answers are listed below, as well as, in the "cover letter-detailed responses to reviewers" and in the revised manuscript ("highlighted manuscript with track changes").

Reviewers’ comments:

Reviewer #1:

The manuscript by Kassaa et al. “High association of COVID-19 severity with poor gut health score in Lebanese patients” is interesting. The manuscript requires minor revision before its publication in PLOS One.

REPLY: We thank you for your time in evaluating the manuscript and your positive response.

Comment 1. The authors avoid the uses of standard deviation in the text and the uses of third person i.e., we etc.

REPLY: As requested, the pronouns “we” and “our” were removed and replaced throughout the text.

Comment 2. Page 3, line 14, the author may add few information about initial prevention approaches (benefits) for COVID-19 i.e. social distancing, improving health (by anti-covid biomolecules), and future challenges by their high mutation ability (variants) and treatments with citations doi: 10.1371/journal.pone.0252300; doi: 10.1007/s12088-020-00893-4; doi: 10.1007/s12088-020-00908-0).

REPLY: As requested, we discussed the benefits of mitigation measures and the challenges related to the emergence of new variants in the respective paragraph.

We added 3 new references to support the information.

Page 5, Lines 1-15

Several mitigation measures have been adopted to prevent the transmission of SARS-CoV-2 and reduce its impact on communities [16]. Non-pharmaceutical prophylactic approaches (such as mask-wearing, washing hands, and physical distancing measures) and COVID-19 vaccines might reduce the amount of virus circulating in and between individuals [17-19]. However, decreasing viral loads might not affect disease severity in secondary cases [20]. Therefore, despite the recent development of effective vaccines against SARS-CoV-2, COVID-19 continues to be problematic, and the probability of exposure to the virus and contracting the disease remains significant across the globe. Notably, vaccinated individuals could potentially still get COVID-19 and transmit live viruses from the upper respiratory tract to others [21]. New variants have rapidly emerged and become dominant worldwide, i.e., the B.1.617.2 (Delta) variant causes more infections and spreads faster than earlier forms of the SARS-CoV-2 [22]. These new variants might also be able to escape from vaccine-induced immunity and other treatment strategies (such as Bamlanivimab/etesevimab). Furthermore, recent data highlighted the strong selective pressure imposed by convalescent plasma therapy, potentially leading to the emergence of SARS-CoV-2 variants with a reduced susceptibility to neutralizing antibodies in immunosuppressed individuals [23].

Comment 3. Page 4, line 19, please add information about correlation of diet, microbiota and COVID (doi: 10.1007/s12088-020-00908-0). Also, add such information in discussion (minor).

REPLY: As requested, we discussed the benefits of mitigation measures and the challenges related to the emergence of new variants in the respective paragraph.

We added 2 new references to support the information.

Page 6, Lines 16-18

Given that diet plays a critical role in modulating the gut microbiota, there has been a serious interest in evaluating the health benefits and disease-preventing properties of diet and dietary habits and their association with a favorable patient outcome [32, 33].

Comment 4. The aim and objectives of this study are not clear.

REPLY: As requested, we clarified the objective of this study.

Page 6, Lines 25-26

Consequently, the current study particularly aimed to demonstrate a potential association between poor gut health and COVID-19 severe symptoms in the Lebanese community.

Comment 5. Fig. 1 quality may be improved (high resolution).

REPLY: As requested, the resolution of figure 1 is improved.

Comment 6. At least one additional Figure (illustration) may be provided as to highlight the summary or prospect of this study.

REPLY: We agree with the reviewer that adding more figures and tables is always a good idea.

However, the nature of our study and findings (multiple metrics and levels of scores) did not allow, despite our attempts, to generate an illustration that might be useful for the readers. We also felt that the illustrations took away from certain important nuances (like flu like illnesses and population properties) highlighted in the study.

 

Reviewer #2: 

This study investigated the possible associations between patients' gut health and COVID-19 disease pathophysiology. It is found that there is a significant association between poor gut health score and the disease symptom severity among Lebanese COVID-19 patients. The results presented in this study is very timely, comprehensive, and pioneering, and would be a big contribution to the fighting against COVID-19 disease if it can be published. This paper would be more compelling if the following minor revision points on gut microbiota and inflammation could be included.

REPLY: We thank you for your time in evaluating the manuscript and your positive response.

Comment 1. In talking about microorganism infection or human microbiota, the human host immunity response has to be taken into consideration. Because of the host immune defenses, most of the viral and bacterial infections are self-limiting. There is no difference with the SARS-COV-2 virus infection, that’s why most of the COVID-19 cases are asymptomatic or mild.

REPLY: We added the requested information.

Page 4, Lines 18-19

Similar to many other microbial infections, mild-to-moderate COVID-19 disease occurs in most patients due to the hosts’ adequate innate and adaptive immune responses [9]. 

Comment 2. Human immune system also has pivotal role in nutrition acquisition from the human microbiota during an acute infection, as both the microorganisms and the damaged host tissue become the source of nutrition with the help of the immunity.

REPLY: This comment is similar to comment 4. As requested, we discussed the association between the nutrition status and the progression of COVID-19 disease.

Page 4, Lines 24-29

Furthermore, previous studies suggested that the nutrition status played a pivotal role in the progression of COVID-19 disease [13, 14]. A hypothesis suggested that hyperinflammation and cytokine storms observed in severe COVID-19 cases are associated with an increase in nutrition acquisition, which may contribute to lipotoxicity and damage in non-adipose tissues, particularly in obese patients or individuals with metabolic syndromes [15].

Comment 3. Although the virulence of the microorganism determines the damages made by the microorganisms to the host cells before they are cleared by host immunity, this microbial or viral virulence alone doesn’t account for the virulence of the disease. The most part of the virulence of an infectious disease is actually the result of the overactive inflammation response of our immune system.

REPLY: As requested, we added a paragraph discussing this comment.

Page 4, Lines 19-24

Given that antiviral immunity is needed to neutralize the virus, inhibit viral replication, and promote the recovery of patients, severe COVID-19 cases may be associated with a dysregulated immune and inflammatory response [10, 11]. High mortality rates were associated with cytokine storms; an excessive production of proinflammatory cytokines that promotes severe acute respiratory distress syndrome and extensive tissue damage, resulting in life-threatening conditions [12].

Comment 4. As a protective response of the human body to remove injurious stimuli and initiate tissue repairing/regeneration process, inflammation is the physiological machinery of immune system to tissue damage. Yet, lipotoxicity as a result of overnutrition will prevent the tissue healing process from happening. Moreover, in the state of over-nutrition, lipotoxicity and tissue damage will form a malignant positive feedback loop, which escalates the inflammation situation and accounts for most of the severity of the COVID-19 disease. So in order to attenuate hyperinflammation, restrictive eating should be performed during infection so that the transient overnutrition from acute infection can be dissipated.

REPLY: This comment is similar to comment 2. As requested, we discussed the association between the nutrition status and the progression of COVID-19 disease.

Page 4, Lines 24-29

Furthermore, previous studies suggested that the nutrition status played a pivotal role in the progression of COVID-19 disease [13, 14]. A hypothesis suggested that hyperinflammation and cytokine storms observed in severe COVID-19 cases are associated with an increase in nutrition acquisition, which may contribute to lipotoxicity and damage in non-adipose tissues, particularly in obese patients or individuals with metabolic syndromes [15].

Comment 5. Illness-induced gastrointestinal symptoms like loss of appetite (illness induced anorexia), diarrhea (cause by certain form of dysbiosis), nausea, vomiting and abdominal pain might be our body’s normal physiological reaction to infection induced over-nutrition, in order to dissipate the nutrition surge of an acute infection. So if these symptoms do not lead to complications like dehydration, they may be beneficial to the host in fighting COVID-19.

REPLY: This information is beyond the objectives of this study; thus, we prefer to not include it in the manuscript.

Comment 6. There are two typos in the manuscript: on page 8, line 5: “Error! Reference source not found.” should be “Figure 1”; 

REPLY: Corrected.

Comment 7. on page 11, line 10, “dry was less commonly reported” should be “diarrhoea was less commonly reported”.

REPLY: Corrected.

Comment 8. The following related references may be included in this review to provide a more complete picture on microbiome, infection, immunity, and inflammation:

Reference:

Azkur AK, Akdis M, Azkur D, Sokolowska M, van de Veen W, Bruggen MC, O'Mahony L, Gao YD, Nadeau K, Akdis CA (2020) Immune response to SARS-CoV-2 and mechanisms of immunopathological changes in COVID-19. Allergy, 75(7):1564-1581. DOI: 10.1111/all.14364

Yu BX, Yu LG & Klionsky DJ. Nutrition Acquisition by Human Immunity, Transient Overnutrition and the Cytokine Storm in Severe Cases of COVID-19. Medical Hypotheses, 155, 110668 (2021). DOI: 10.1016/j.mehy.2021.110668.

Papa A, Covino M, Pizzolante F, Miele L, Lopetuso LR, Bove V, Iorio R, Simeoni B, Vetrone LM, Tricoli L, Mignini I, Schepis T, D'Alessandro A, Coppola G, Nicoletti T, Visconti E, Rapaccini G (2020) Gastrointestinal symptoms and digestive comorbidities in an Italian cohort of patients with COVID-19. Eur Rev Med Pharmacol Sci 24:7506-7511. DOI: 10.26355/eurrev_202007_21923

Troisi J, Venutolo G, Pujolassos Tanyà M, Delli Carri M, Landolfi A, Fasano A. COVID-19 and the gastrointestinal tract: Source of infection or merely a target of the inflammatory process following SARS-CoV-2 infection? World J Gastroenterol 2021; 27(14): 1406-1418. DOI: 10.3748/wjg.v27.i14.1406

REPLY: All suggested references were included.

---

## [Editor Report · Decision Letter 1]

8 Oct 2021

High association of COVID-19 severity with poor gut health score in Lebanese patients

PONE-D-21-25958R1

Dear Dr. Osman,

We’re pleased to inform you that your manuscript has been judged scientifically suitable for publication and will be formally accepted for publication once it meets all outstanding technical requirements.

Kind regards,

Sanjay Kumar Singh Patel, Ph.D.

Academic Editor

PLOS ONE

---

## [Editor Report · Acceptance letter]

14 Oct 2021

PONE-D-21-25958R1 

High association of COVID-19 severity with poor gut health score in Lebanese patients 

Dear Dr. Osman:

I'm pleased to inform you that your manuscript has been deemed suitable for publication in PLOS ONE. Congratulations! Your manuscript is now with our production department. 

Kind regards, 

on behalf of

Dr. Sanjay Kumar Singh Patel 

Academic Editor

PLOS ONE